# Porcine Reproductive and Respiratory Syndrome Virus: Challenges and Advances in Vaccine Development

**DOI:** 10.3390/vaccines13030260

**Published:** 2025-02-28

**Authors:** Zhan He, Fangfang Li, Min Liu, Jiali Liao, Chunhe Guo

**Affiliations:** Guangdong Laboratory for Lingnan Modern Agriculture, State Key Laboratory for Animal Disease Control and Prevention, Key Laboratory of Zoonosis Prevention and Control of Guangdong Province, College of Veterinary Medicine, South China Agricultural University, Guangzhou 510642, China; hezhan1021@stu.scau.edu.cn (Z.H.); 20234073011@stu.scau.edu.cn (F.L.); liumin0110@stu.scau.edu.cn (M.L.); liaojiali@stu.scau.edu.cn (J.L.)

**Keywords:** PRRSV, immunogenicity, vaccine, protection effect

## Abstract

Persistent infection of porcine reproductive and respiratory syndrome virus (PRRSV) significantly hampers both the quantity and quality of pork production in China. Although PRRSV is widely prevalent worldwide, the absence of effective vaccines has made it one of the major pathogens threatening the sustainable development of the global swine industry. Vaccination remains one of the most effective measures for controlling pathogen infections. However, the continuous genetic recombination and mutation of PRRSV demand more comprehensive strategies to address emerging threats, while ensuring the efficacy and safety of vaccines. This review provides an overview of the latest advances in PRRSV vaccine research, highlighting the importance of understanding the unique strengths and limitations of various vaccines in developing effective therapeutic approaches and vaccination strategies. Moreover, the development of adjuvants and antiviral drugs as adjuncts to combat PRRSV infection offers significant potential for enhancing disease control efforts. With the advancement of technologies such as proteolysis-targeting chimera (PROTAC) and mRNA, new avenues for controlling PRRSV and other pathogens are emerging, offering considerable hope. Ultimately, the goal of these vaccine developments is to alleviate the impact of PRRSV on animal health and the profitability of the swine industry.

## 1. Introduction

Porcine reproductive and respiratory syndrome virus (PRRSV) presents a significant challenge to the global swine industry. The virus’s high propensity for recombination and mutation facilitates the emergence of diverse strains, such as HP-PRRSV, NADC30, and JXA1, which exacerbate infection risks and complicate control efforts. Since the emergence of the CH-1a strain in China in 1995, the swine industry has experienced multiple outbreaks, including the highly pathogenic HP-PRRSV in 2006 and the NADC30 variant in 2013. These outbreaks have severely hindered the sustainable development of China’s swine industry over the past three decades (Figure 1A). PRRSV is an enveloped, single-stranded, positive-sense RNA virus belonging to the genus Arterivirus within the Arteriviridae family [1]. The virus primarily causes reproductive disorders in pregnant sows and respiratory diseases in pigs of all ages [2]. Transmission occurs through direct contact with excretions or secretions from infected pigs, and potential airborne or avian-mediated transmission further complicates disease management [3,4].

PRRSV is classified into two species: PRRSV-1 (European type) and PRRSV-2 (North American type) [4,5]. Although both species cause similar clinical manifestations, they exhibit significant genetic differences. In China, PRRSV-2 has been the predominant strain; however, a 2016 study in Guangdong Province revealed a PRRSV-1 positivity rate of 24.8%, highlighting its increasing prevalence [6]. PRRSV-1 has since spread to 23 provinces, municipalities, and autonomous regions across China, including central, northern, southern, eastern, northeastern, and southwestern areas [7]. While most PRRSV-1 infections are associated with milder clinical symptoms and lower pathogenicity, its potential threat cannot be ignored given the virus’s genetic diversity. A study by Shi et al. demonstrated that phylogenetic analysis based on the ORF5 sequence classifies PRRSV-2 into nine lineages and 37 sublineages, with the predominant strains in China primarily distributed in Lineages 1, 3, 5, and 8 [8]. Our epidemiological survey reveals that, in recent years, the PRRSV strains circulating in China are predominantly Lineage 1, accounting for over 70% of cases. Among these, sublineage 1.8 (NADC30-like) is currently the most prevalent, representing 65.20%, while sublineage 1.5 (NADC34-like) emerged gradually after 2017 and now constitutes a growing proportion (8.80%), potentially becoming a major strain in the future. Notably, the spread of Lineage 1 strains in China has accelerated significantly, and previous studies have indicated a higher propensity for recombination mutations both within and between lineages, which may enhance the virus’s adaptability and immune escape, making control efforts more challenging [9]. Additionally, Lineage 8 (CH-1a-like and HP-PRRSV-like) strains remain prevalent in China (18.16%), with HP-PRRSV-like strains having caused significant economic losses since the 2006 outbreak and continuing to show a stable presence. These strains exhibit strong genetic stability and warrant continued attention. In contrast, Lineage 3 (QYYZ-like) and Lineage 5 (BJ-4-like) strains are less frequently detected (5.16% and 2.68%, respectively), but their potential evolutionary and epidemiological risks should not be overlooked (Figure 1B,C).

The extensive genetic diversity of PRRSV is largely driven by its high recombination rate, which promotes the emergence of new strains with enhanced immune evasion capabilities [10]. For example, recombination between NADC30 and HP-PRRSV strains, particularly in the nsp2 region, has resulted in variants with increased pathogenicity, underscoring the complex interplay between viral evolution and host immunity [11]. These dynamic genetic changes present significant challenges for vaccine development and underscore the importance of closely monitoring viral evolution to mitigate potential risks.

Vaccination remains the most effective approach for controlling PRRSV, aiming to enhance both humoral and cellular immune responses while reducing morbidity and mortality [12]. Advances in vaccine platforms, including RNA and subunit vaccines, offer promising avenues for improved protection [13]. Adjuvants, which play a pivotal role in enhancing vaccine efficacy, are being developed in parallel to maximize immunogenicity [14]. Additionally, antiviral agents provide complementary strategies for PRRSV control [15]. However, the continuous evolution and recombination of PRRSV highlight the necessity of designing adaptive and effective vaccines to address emerging threats. Consequently, the development of effective vaccines is crucial not only for controlling PRRSV, but also for ensuring the stability of China’s pig farming sector and safeguarding food security.

This study provides a comprehensive overview of recent advancements in the development of vaccines targeting PRRSV, emphasizing the strengths and limitations of existing vaccine platforms. By exploring strategies for the development of adjuvants and antiviral agents, the research offers valuable insights into improving PRRSV control and prevention. Ultimately, this research aims to mitigate the impact of PRRSV on animal health and promote the sustainability of swine production systems.

## 2. Types of PRRSV Vaccines

Commercially available PRRSV vaccines primarily consist of killed virus (KV) and modified live virus (MLV) vaccines (Table 1). While KVs offer excellent safety and stability, with no risk of reversion to virulence, their immunogenicity is suboptimal [9,16,17]. They fail to effectively stimulate PRRSV-specific antibodies and cell-mediated immunity (CMI), often requiring potent adjuvants and multiple doses to achieve adequate immune responses. This highlights that KV vaccines function more as therapeutic vaccines than as preventative measures against the disease.

MLV vaccines are the most widely used, and early studies indicated that MLV vaccines provided some protection against genetically homologous wild-type PRRSV strains. However, they have several notable limitations: (1) incomplete protection against heterologous strains [16], (2) potential recombination with wild-type PRRSV or other MLVs, resulting in vaccine-derived PRRSV strains [18,19], (3) risks of virulence reversion and the poorly understood phenomenon of antibody-dependent enhancement (ADE) [20]. Both KVs and MLVs derived from killed or modified live PRRSV have limited efficacy against the highly diverse circulating strains of PRRSV-1 and PRRSV-2. Therefore, there is an urgent need to develop more effective and broadly protective PRRSV vaccines.

To overcome the shortcomings of first-generation vaccines, second-generation vaccines focus on subunit components, including subunit antigens, conjugated/recombinant antigens, or synthetic proteins. Recombinant subunit vaccines do not use the whole virus (live or inactivated); instead, they rely on antigen expression and purification. Plana et al. conducted animal immunization tests using PRRSV ORF2-ORF7 expression vectors, demonstrating that GP3 and GP5 can provide specific protective effects. This suggests that GP3 and GP5 subunit vaccines possess good immunogenicity and may serve as promising candidates for the development of recombinant subunit vaccines [9]. In the long term, PRRSV subunit vaccines hold significant promise for PRRSV control. However, the lack of pathogen-associated molecular patterns (PAMPs) in subunit vaccines limits their ability to induce strong immune responses, necessitating the use of adjuvants to enhance their efficacy [21]. Given the crucial role of adjuvants in boosting antigen immune responses, there is a parallel need to develop adjuvants that can enhance the specific immune responses of PRRSV subunit vaccines.

Third-generation vaccines include gene-based platforms, such as DNA and RNA vaccines, viral vector platforms, and chimeric vaccines that combine live or inactivated components. In recent years, third-generation vaccines have demonstrated immense potential in PRRSV vaccine research. As vaccine development technologies continue to advance, these innovative approaches have had a significant impact on animal health and pig production. Despite some remaining challenges, we believe that with the progress of biotechnology, the development of third-generation vaccines may become a focal point in the PRRSV vaccine field (Figure 2).

## 3. Efficacy and Challenges of First-Generation Vaccines

In China, two widely used commercial MLVs, Ingelvac PRRS^®^ MLV and Ruilanan^®^, are derived from the classic PRRSV strain VR2332 and the highly pathogenic (HP) PRRSV strain TJM-F92, respectively. Both vaccines offer significant protection against homologous and heterologous strains, such as NADC30, effectively reducing clinical disease incidence and severity, improving growth performance, and shortening the duration of viremia and viral shedding [22,23]. However, the VR2332-based vaccine shows limited cross-protection against HP-PRRSV strains, such as TP, whereas TJM-F92 and JXA1-R vaccines provide better protective efficacy [24]. Further studies reveal that the TJM-F92 vaccine only partially protects against heterologous PRRSV-NADC30 strains. This limitation is attributed to the vaccine’s induction of a significant expansion of CD8^+^ T cells without effectively stimulating CD4^+^ T cells or γδ T cells [25].

Since 2013, NADC30-like PRRSV has become the predominant strain in China, while existing commercial vaccines provide limited protection against it [26,27]. Researchers have addressed this gap by passaging the NADC30-like strain SD 125 times in Marc145 cells, yielding an attenuated PRRSV strain, SD-R [28]. The SD-R vaccine demonstrates safe and effective protection against homologous and heterologous NADC30-like strains [28]. In another study, the SD-R vaccine also protected against HP-PRRSV challenges, reducing thymic atrophy and lung consolidation caused by the HP-PRRSV HuN4 strain [29].

PRRSV-A2MC2, a moderately virulent strain with the highest similarity to the prototype strain VR-2332, is notable as the first PRRSV strain reported to induce interferon (IFN) synthesis [30]. Compared to pigs infected with MLV or the virulent VR-2385 strain, A2MC2 infection results in enhanced neutralizing antibody production. Attenuated A2MC2 effectively protects against the VR-2385 challenge, demonstrating its potential as a vaccine candidate. However, when used against highly virulent strains such as MN184, A2MC2’s protective efficacy diminishes, likely due to recombination events between PRRSV strains [31].

The use of MLV vaccines has been linked to the rapid evolution of PRRSV. These vaccines impose selection pressures that can drive the emergence of novel strains. PRRSV’s low replication fidelity, combined with the replication capability of MLV strains within the host, contributes to viremia, viral shedding, and risks of virulence reversion or recombination with field strains, potentially resulting in more virulent variants [32,33]. Studies indicate that attenuated PRRSV strains may regain virulence through serial passaging, enhancing their adaptation to porcine alveolar macrophages and replication efficiency [34]. Moreover, MLV strains, like wild-type PRRSV, have been reported to cause immune dysregulation in piglets [35].

Compared to MLV vaccines, KV vaccines have been approved globally due to their higher safety profile. The protective immune response induced by most KV vaccines is mediated by enhanced viral neutralizing (VN) antibody responses [13]. VN antibodies are responsible for clearing the virus from the lungs and reducing PRRSV transmission via the placenta, highlighting their crucial role in protective immunity [36]. Given the significant role of PRRSV VN antibodies in viral clearance in pigs, research on PRRSV KV vaccines should focus on inducing protective VN antibody responses.

## 4. Advances in Subunit Vaccine Development for PRRSV

Although first-generation vaccines have been highly effective against certain infectious diseases, they fail to provide complete protection against PRRSV infection. Moreover, concerns persist due to the inability of these vaccines to fully eliminate the risk of recombinant viruses. Subunit vaccines that utilize microbial fragments may overcome these challenges. Unlike first-generation vaccines, subunit vaccines (Table 2) contain only the pathogen-specific antigen components necessary to induce an appropriate immune response, typically found on the surface of the virus. These vaccines do not contain live microorganisms, which prevent reversion to infectious or toxic forms, making them suitable for use in infected pigs or immunocompromised hosts. As a result, adverse immune reactions following vaccination are significantly reduced [37]. Thus, subunit vaccines are considered a safer alternative. However, due to their non-replicating nature and limited components, their immunogenicity is relatively weak. To enhance their efficacy, subunit vaccines require higher doses, booster shots, and the co-administration of immune stimulators (adjuvants). To induce effective immunity against PRRSV, exposing immunogenic viral proteins in the form of “heterodimers and heterotetramer complexes” is crucial for efficiently inducing specific and protective B cell and T cell responses in innovative vaccine formulations.

Immunogenic PRRSV proteins exist in both structural and non-structural forms. Among the PRRSV structural proteins, GP3, GP5, and M have been extensively studied for their potential in subunit vaccines [38,39]. Ma et al. constructed three recombinant adenoviruses—rAd-GP5, rAd-M, and rAd-M-GP5—using a replication-deficient rAd5 vector system. Immunization with rAd-M-GP5 induced higher neutralizing antibody titers and stronger anti-PRRSV cytotoxic T lymphocyte (CTL) responses compared to the other two groups [40]. Similarly, Wang et al. used rAd-GP3-GP4-GP5 to induce higher neutralizing antibody titers and stronger CTL responses against PRRSV [41]. These findings further support the development of adenovirus-based recombinant vaccines targeting PRRSV structural proteins.

The development of PRRSV subunit vaccines currently involves various approaches, including nanoparticle delivery systems, DNA vaccines, and recombinant subunit vaccines produced using baculovirus, plant, or replication-deficient viral vectors. Notably, the recombinant PRRSV vaccine rPRRSV-E2 has shown promise as a viral vector-based live vaccine, offering 100% protection against highly pathogenic PRRSV (HP-PRRSV) and the classical swine fever virus (CSFV) [42]. Using reverse genetics, Gao et al. developed rPRRSV-E2, which expresses the E2 protein of CSFV. Immunization with rPRRSV-E2 induced long-lasting antibody responses against both PRRSV and CSFV for up to 23 weeks, with all vaccinated pigs providing 100% protection against HP-PRRSV or CSFV challenges even 24 weeks post-vaccination [43].

T cells are crucial for controlling viral infections through cytolysis and cytokine secretion. In the absence of neutralizing antibodies, T cells play a key role in controlling PRRSV infection. Potential T cell epitopes in PRRSV are present in proteins such as GP3, GP4, GP5, M, N, nsp2, nsp5, nsp9, and nsp10. Herpesvirus vectors have been shown to enhance T cell responses to heterologous target antigens, particularly boosting CD8^+^ T cell activity [44]. The M and nsp5 proteins of PRRSV-1 contain conserved major epitopes for CD4^+^ and CD8^+^ T cells, which are recognized in multiple PRRSV strains and inbred pigs [45,46]. A vaccine based on the herpesvirus vector BoHV-4-M-nsp5, which expresses fusion proteins of M and nsp5, induced specific CD4^+^ and CD8^+^ T cell responses in the bloodstream. Following priming and boosting, the enhanced expression of IFN-γ and TNF was observed. However, this vaccine was not effective in controlling PRRSV infection, although it did alleviate pulmonary pathology [47].

Using nanoparticles (NPs) for multivalent antigen delivery is an effective strategy to enhance the immunogenicity of subunit vaccines and induce robust immune responses [48]. Ferritin has been widely utilized as a nanoparticle carrier for developing viral vaccines. Ma et al. fused modified GP5 proteins with ferritin (GP5m-Ft) using a baculovirus system to produce GP5m-Ft nanoparticles. The GP5m-Ft subunit vaccine induced specific protective immune responses and significantly enhanced Th1-type cellular immunity [49]. Building on this, Chang et al. used *E. coli* to express the GP5m-Ft protein at high levels, achieving similar protective outcomes, including reduced fever, viremia, and lung lesion scores in vaccinated pigs [50]. The genetic variability of PRRSV poses challenges for the vaccine coverage of regional circulating strains. Multiepitope vaccine designs aim to address this by targeting diverse strains. For example, a novel chimeric protein (LTB-PRRSV) was developed, fusing multiple GP5 epitopes with the heat-labile enterotoxin B subunit (LTB) from *E. coli*. Under various delivery systems, LTB-PRRSV induced strong humoral responses, and when delivered with layered double hydroxides (LDH), it not only extended the duration of the humoral response but also promoted anti-inflammatory responses associated with IL-6 and TNF-α secretion, while reducing platelet aggregation and leukocyte migration [51]. Thus, LDH has also been considered a potential adjuvant for veterinary vaccines.

Production cost is a critical factor for livestock farmers, and to minimize vaccine production and design costs, some researchers have started exploring plant expression systems for developing veterinary vaccines. For example, An et al. expressed codon-optimized, transmembrane-deleted recombinant PRRSV GP4D and GP5D glycoproteins in Arabidopsis thaliana, achieving effective immune responses through oral administration. This immunization increased the PRRSV-specific antibody titers and levels of pro-inflammatory cytokines (TNF-α and IL-12) [52]. In another study, piglets fed transgenic banana leaves expressing GP5 showed vaccine-dependent increases in anti-PRRSV IgG and IgA levels in both serum and saliva [53]. Similarly, GP5 expression in tobacco leaves induced both mucosal and systemic humoral and cellular immune responses [54]. Recent advancements in transient expression allow plant-based systems to rapidly scale up the production of recombinant proteins, and due to the edibility of plant tissues, they avoid the need for expensive protein purification, significantly lowering costs [55]. Moreover, oral administration can directly target intestinal mucosa, enhancing IgA titers [56]. Unlike parenteral administration, which often results in higher IgG levels and lower IgA levels, oral vaccines can stimulate higher IgA production in intestinal mucosal tissues, leading to increased IgA levels in other mucosal sites, such as the respiratory tract [57]. These findings underscore the potential of plant expression systems as a viable platform for developing subunit vaccines to control PRRSV infections (Figure 3).

**Table 2 vaccines-13-00260-t002:** The study of PRRSV subunit vaccines.

Vaccine	Expression System	Method	Results and References
rAd-M-GP5	Adenovirus	Co-expression	Induces higher neutralizing antibody titers and stronger anti-PRRSV CTL responses [40].
rAdGP3-GP5-GP4	Adenovirus	Co-expression	Produces high levels of PRRSV neutralizing antibody titers [41].
rPRRSV-E2	PRRSV-based vector	Insertion of the E2 gene of CSFV between ORF1b and ORF2 in the genome of PRRS vaccine virus HuN4-F112	Offers 100% protection against highly pathogenic PRRSV and CSFV [42].
BoHV-4-M-nsp5	Herpesvirus-based vector	Uses a synthetic helical linker	Induces specific CD4+ and CD8+ T cell responses, and reduces lung lesions but does not effectively control PRRSV infection [47].
GP5m-Ft	Baculovirus	Uses a Gly-Gly-Gly-Ser linker	Enhances Th1-type cellular immunity [49].
GP5m-Ft	*Escherichia coli*	Uses a Gly-Gly-Gly-Ser linker	Reduces fever, viremia, and lung lesion scores in vaccinated pigs [50].
LTB-PRRSV	LDH delivery	Connects the four selected epitopes of GP5 in series with the LTB via the GPGP connector	Induces a strong humoral response and prolongs its duration [51].
GP4D/GP5D	Plant	Deletion of transmembrane regions and codon optimization for expression in Arabidopsis thaliana	Increases PRRSV-specific antibody titers and levels of proinflammatory cytokines (TNF-α and IL-12) [52].
GP5	Plant	Expression of GP5 in banana	Increases in anti-PRRSV IgG and IgA in serum and saliva [53].
GP5-T	Plant	Expression of GP5 in tobacco	Oral immunization induces mucosal and systemic humoral and cellular immune responses [57].

## 5. mRNA Vaccines Are an Attractive Alternative to Traditional Vaccines

mRNA technology offers several advantages, making it an attractive alternative to traditional and even DNA-based vaccines. Unlike live-attenuated or inactivated vaccines, mRNA provides precision by encoding only specific antigens, thereby inducing a targeted immune response. Additionally, it stimulates both humoral and cellular immunity while activating the innate immune system [58]. Compared to DNA vaccines, mRNA demonstrates higher efficacy and safety, with virtually no risk of random genomic integration [59]. Moreover, antigen expression from mRNA is transient, as the molecule is rapidly degraded by host cells within 2–3 days [60].

The SARS-CoV-2 pandemic highlights the effectiveness of mRNA vaccines. In 2021, the first COVID-19 mRNA vaccine, developed by Pfizer-BioNTech, received full approval [61]. Currently, mRNA vaccines are being explored for various veterinary applications, including research on foot-and-mouth disease, pseudorabies, and porcine epidemic diarrhea [62,63,64]. Compared to traditional inactivated and attenuated vaccines, mRNA vaccines offer significantly shorter preparation times, exceptional safety profiles, and prolonged efficacy. The rapid development of mRNA vaccines has sparked great optimism for controlling and preventing diseases like PRRS on a global scale.

Upon delivery into cells, mRNA vaccines trigger the short-term production of antigens, which aggregate in the cytoplasm [65]. Once inside the cell, the mRNA is translated directly by ribosomes into proteins (antigens). These proteins are then processed by the proteasome and transported to the surface of the cell via MHC-I molecules. The antigen peptides presented by MHC-I molecules activate CD8^+^ T cells. Meanwhile, viral proteins that are secreted or membrane-anchored are detected by B cells, initiating an antibody-mediated response against the specific antigen. Furthermore, after antigen peptides are engulfed by antigen-presenting cells (APCs), they are processed by MHC-II molecules, which activate CD4^+^ T cells. This CD4^+^ T cell response plays a crucial role in initiating the antibody-mediated response [66].

Luo et al. developed two mRNA vaccines targeting the HP-PRRSV strain in China based on a non-replicating mRNA vaccine expression platform: GP5-mRNA and GP2-GP5-M-mRNA [67]. The team synthesized plasmids carrying GP5 and GP2-GP5-M mRNAs, purifies them, and transcribes the mRNA in vitro. The mRNA was then transfected into 293T cells for expression verification. Subsequently, lipid nanoparticles (LNPs) are used to encapsulate mRNA and form the complete mRNA vaccine. Immune evaluation in mice revealed that the GP5-mRNA vaccine induces significantly higher levels of GP5 antibodies and neutralizing titers than the GP2-GP5-M-mRNA vaccine. Additionally, it stimulates stronger CD8^+^ T cell responses, with higher levels of IFN-γ, TNF-α, and IL-4 cytokines, indicating a robust cellular immune response [67].

Self-amplifying RNA (SaRNA) vaccines are an emerging platform that uses the replication machinery of viruses, such as the Venezuelan equine encephalitis (VEE) virus or Sindbis virus (SIN), for RNA self-amplification [68]. Compared to conventional mRNA vaccines, SaRNA vaccines provide equivalent protection at significantly lower doses [69]. However, SaRNA molecules, typically over 8000 nucleotides, are much larger than standard mRNA and more challenging to deliver into cells [70,71,72]. Similar to mRNA vaccines, SaRNA vaccines require lipid nanoparticles, liposomes, or multiple nanomicelles for cellular delivery. Chang et al. utilized a VEE-based SaRNA platform to express PRRSV antigens, specifically the M protein (SaRNA-M) and the dNGP5 protein (SaRNA-dNGP5, a modified version of GP5 with a reduced immunogenic N-glycosylation site) [73]. SaRNA-M and SaRNA-dNGP5 were expressed in transfected cells for at least 28 days. Notably, the expression level of the M protein was significantly lower than that of GP5, possibly due to M protein accumulation in the ER membrane, which restricts its expression. Immune piglet assessment shows that SaRNA-dNGP5 induces stronger neutralizing antibodies and elicits more potent IL-4 and IFN-γ responses, providing improved protection against homologous and heterologous PRRSV strains [73]. Currently, the main challenges associated with RNA vaccines are their instability and inability to penetrate physiological barriers, preventing them from reaching target cells [69]. In the body, mRNA is prone to degradation by RNases or recognition and phagocytosis by macrophages or dendritic cells (DCs) in the liver [70]. While naked mRNA can still be absorbed by cells, the process is inefficient, requiring repeated administration [69]. However, excessive dosing can lead to immune-related adverse reactions.

## 6. Adjuvants Are of Key Value in Vaccine Development

To address the problem of poor immune effects, the design and selection of adjuvants are crucial. Adjuvants are typically used to enhance immune responses in vaccines. Those that induce both innate and adaptive immunity, as well as systemic and mucosal responses, can significantly improve the efficacy of PRRSV vaccines. They help enhance the body’s immune response by precisely targeting and reducing the overall volume of the drug, optimizing drug delivery routes, lowering toxicity, and boosting therapeutic efficacy. Common adjuvants include cytokines (e.g., IL-4, IL-15, IL-18, CD40, and granulocyte-macrophage colony-stimulating factor [GM-CSF]), bacterial products (e.g., cholera toxin, HSP70), and chemical agents (e.g., lipopolysaccharide, Poly IC-LC) [74,75,76,77]. Table 3 mentions the adjuvants used in the development of PRRSV vaccines.

Astragaloside IV (AS-IV), a saponin compound, has been shown to activate the cGAS-STING signaling pathway, thereby inducing IFN production and mitigating the immunosuppressive effects caused by PRRSV infection [78]. Studies examining PAM cells reveal that PRRSV infection induces immune dysfunction and impairs antigen-presenting capacity in these cells. However, treating AS-IV alleviates these impairments, highlighting the potential value of immune system stimulants in mitigating PRRSV-induced immunosuppression.

Heat shock proteins (HSPs), which serve as intracellular chaperones in mammalian cells and microorganisms, are highly conserved [79]. Studies have shown that HSPs can act as potent immune adjuvants, enhancing innate and adaptive immune responses [74]. For example, Li et al. fused HSP70 with PRRSV GP3/GP5 and expressed it in a recombinant adenovirus (rAd-HSA35). Immunization of piglets with this recombinant virus results in significant increases in serum IFN-γ and IL-4 levels and alleviates viremia and lung pathology caused by PRRSV infection [80]. TLR agonists, which activate the host immune systems, have also been shown to promote dendritic cell maturation and induce proinflammatory cytokine production (such as TNF-α, IL-1β, IL-6, and IFN-β). These responses subsequently activate cytotoxic T lymphocytes [81]. HSP70c and HSPX have been identified as strong agonists of TLR2 and TLR4. In another study, HSP70c and HSPX, when used as adjuvants, significantly enhanced the protective efficacy of PRRSV vaccines. When these proteins are expressed in *Escherichia coli* and purified, and then co-administered with a CPD attenuated chimeric PRRSV through intramuscular or intradermal injection into 3-week-old pigs, the results show elevated levels of PRRSV-specific IFN-γ, enhanced cytokine responses, and a significant reduction in lung pathology and viral load [82].

GM-CSF can expand dendritic cell (DC) populations, enhancing humoral and cellular immune responses and influencing the Th1/Th2 cytokine balance [83,84]. GM-CSF is widely used as an effective adjuvant to boost vaccine-induced immune responses [85]. For example, Wang et al. fused GM-CSF with PRRSV GP3/GP5, and this fusion protein significantly increased the secretion of IFN-γ and IL-4 in both mouse lymphocyte cultures and pig serum, thereby enhancing humoral and cellular immunity and providing protection against PRRSV [86]. Additionally, when pig IFN-γ (poIFN-γ) and GM-CSF were co-expressed and combined with PRRSV KV for immunization, there was a notable increase in neutralizing antibody titers, accelerated viral clearance, alleviation of clinical symptoms, and protection against highly pathogenic PRRSV infections [76]. Another cytokine, CD40 ligand, has been shown to effectively enhance humoral and cell-mediated immune responses to PRRSV GP3 and GP5 [75].

Given the strong performance of cytokines and co-stimulatory factors in enhancing vaccine immune responses, several studies have explored the use of cytokines as vaccine adjuvants to improve PRRSV vaccine efficacy. For instance, Cao et al. fused the porcine cytokines IL-15 or IL-18 with a membrane-targeting signal from CD59 and incorporated them into a recombinant PRRSV MLV. This approach significantly enhances NK cell and γδ T cell responses and provides enhanced protection against the PRRSV NADC20 strain [77]. In another study, a recombinant adenovirus vaccine expressing GP3 and GP5 from the European LV strain (M96262), fused with IL-18, was evaluated with Quil A as an adjuvant. This combination produces higher levels of GP3- and GP5-specific antibodies, with neutralizing antibody titers greater than 1:16. The fusion of IL-18 significantly boosts IFN-γ and IL-4 secretion in pig serum upon PRRSV stimulation [87]. Furthermore, porcine IL-4 (a Th2-type cytokine) has significantly enhanced the protective immune responses of PRRSV MLV vaccines [88]. Research indicates that the novel IgM-type monoclonal antibody PR5nf1 (Mab-PR5nf1), which targets both PRRSV-1 and PRRSV-2, can recognize epitopes on intact PRRSV particles [89]. When used as an adjuvant in an inactivated vaccine, it enhances the production of PRRSV-specific CTLs, addressing the deficiency of CTL production commonly observed in traditional KV vaccines. Immunization experiments in piglets show that the inclusion of Mab-PR5nf1 as an adjuvant increases serum-neutralizing antibody levels, reduces viral shedding, improves overall survival rates, and boosts cell-mediated immunity, enhancing protection against heterologous HP-PRRSV challenges [90].

Several studies have also highlighted the significant antiviral activity of herbal extracts at various stages of the PRRSV life cycle. For instance, epigallocatechin-3-gallate (EGCG) can block PRRSV adsorption, while curcumin and tetrahydroaltersolanol C (TD-C) interfere with PRRSV internalization [91,92,93]. Other compounds, such as sodium tanshinone IIA sulfonate (STS) and flavaspidic acid AB (FA-AB), inhibit viral replication, and pyrithione (PT) prevents the assembly and release of PRRSV particles [94,95,96]. Some herbal extracts have been used as adjuvants alongside PRRSV MLV or KV vaccines to enhance their immune efficacy. Quil A, known for stimulating Th1 immune responses and promoting the generation of cytotoxic T lymphocytes, has enhanced the immune responses of vaccines against Actinobacillus pleuropneumonia, influenza viruses, and the foot-and-mouth disease virus [97,98,99]. When Quil A is used as an adjuvant with the PRRSV-1 MLV vaccine in piglets, it upregulates Type I interferon-regulating genes, enhances the expression of both Type I and II interferons, and increases the production of proinflammatory cytokines, leading to effective cross-protection and reduced viremia caused by PRRSV-2 [100].

In another study, oral administration of Houttuynia cordata extract (HC) enhanced cross-protection from PRRSV-1 MLV vaccines, significantly boosting the expression of interferon-regulating genes and both Type I and II interferons in PBMCs from pigs infected with HP-PRRSV-2 [101]. Quercetin was found to improve the cross-protection of PRRSV-1 MLV vaccines against HP-PRRSV-2 by significantly increasing IRG expression and both interferon types, while reducing proinflammatory and anti-inflammatory cytokine expression in macrophages from HP-PRRSV-infected pigs [102]. Additionally, astragalus (AM) and curcumin (CZ), known for their anti-inflammatory, immunostimulatory, antibacterial, and anti-tumor effects, have been shown to have complementary synergistic effects when used together [103]. Research has demonstrated that a combination of AM and CZ extracts can inhibit PRRSV replication in vitro, reduce the secretion of proinflammatory cytokines such as IL-1β and TNF-α, and upregulate IFN-α mRNA expression. Supplementation with AM and CZ extracts also increases PRRSV-specific antibody levels in piglets, alleviating inflammation by reducing proinflammatory cytokine secretion and enhancing IL-4 and IL-2 levels [104].

**Table 3 vaccines-13-00260-t003:** The study of PRRSV vaccine adjuvants.

Adjuvant	Antigen or Virus Strain	Results and References
HSP70	GP3, GP5	Increases in serum IFN-γ and IL-4 levels and alleviates viremia and lung pathology caused by PRRSV infection [80].
HSP70c and HSPX	LMY-BP-CPD	Enhances PRRSV-specific IFN-γ levels and reduces lung pathology and viral load significantly [82].
GM-CSF	GP3, GP5	Increases the secretion of IFN-γ and IL-4 in both mouse lymphocyte cultures and pig serum [86].
IFN-γ (poIFN-γ) and GM-CSF	PRRSV KV	Increases neutralizing antibody titers, accelerates viral clearance, and protects against highly pathogenic PRRSV infections [76].
CD40	GP3, GP5	Enhances GP3 and GP5-mediated humoral and cellular immune responses [75].
IL-15	PRRSV MLV	Enhances NK cell and γδ T cell responses and protection against the PRRSV NADC20 strain [77].
IL-18 and Quil A	GP3, GP5	Produces neutralizing antibody titers greater than 1:16 and boosts IFN-γ and IL-4 secretion [87].
IL-4	PRRSV MLV	Increases the ratio of CD3^+^CD4^+^/CD3^+^CD8^+^ T-lymphocyte subpopulations in attacked piglets and alleviates clinical symptoms [88].
Mab-PR5nf1	PRRSV KV	Enhances the production of PRRSV-specific CTLs, boosts cell-mediated immunity, and enhances the protection against heterologous HP-PRRSV challenges [90].
Quil A	PRRSV MLV	Enhances the expression of both Type I and II interferons [100].
HC	PRRSV MLV	Boosts the expression of interferon-regulating genes and both Type I and II interferons in PBMCs from pigs infected with HP-PRRSV-2 [101].
Quercetin	PRRSV MLV	Increases IRG expression and both interferon types, reduces proinflammatory and anti-inflammatory cytokines expression in macrophages from HP-PRRSV-infected pigs [102].
AM and CZ	PRRSV KV	Inhibits PRRSV replication in vitro, reduces the secretion of pro-inflammatory cytokines (e.g., IL-1β and TNF-α), and increases the expression of IFN-α mRNA [104].

## 7. Antiviral Drugs/Antibodies Targeting the PRRSV Life Cycle Are a Compensatory Approach to Mitigate PRRSV Infection

PRRSV exhibits strict cellular and tissue tropism. Apart from porcine alveolar macrophages (PAMs), PRRSV has also been shown to replicate in the porcine testicular respiratory epithelial cells and spermatogenic cells. During the viral infection process, PRRSV enters host cells through receptor-mediated endocytosis. Specifically, the heterodimer of GP5 and M proteins interacts with the N-terminal domains of heparan sulfate (HS) and sialic acid-binding immunoglobulin-like lectin (Sn or CD169), while GP4 interacts with heat shock protein family A member 8 (HSPA8), and both facilitate PRRSV internalization [105]. As a chaperone protein, HSPA8 plays an essential physiological role in protein folding and degradation, primarily dependent on its two functional domains: the amino-terminal ATPase domain (1–383 aa, referred to as the AB domain) and the carboxyl-terminal peptide-binding domain (393–646 aa, referred to as the PB domain). In the context of PRRSV infection, the interaction between GP4 and HSPA8 contributes to virus particle adhesion on the cell surface. It promotes its internalization through clathrin-mediated endocytosis (CME), a process closely tied to the ATPase activity of HSPA8 [106].

CD163 plays a pivotal role in the PRRSV lifecycle by facilitating viral uncoating and the release of the genome from early endosomes into the cytoplasm through interactions with viral envelope glycoproteins [107]. CD163 is a transmembrane protein belonging to the cysteine-rich scavenger receptor (SRCR) superfamily that is essential for PRRSV infection. The CD163 gene comprises 17 exons that encode a signal peptide and nine tandem SRCR domains (SRCR1 to SRCR9) [108]. Among these, Guo et al. found that the deletion of SRCR5 did not affect the co-localization of CD163 with the PRRSV-N protein in early endosomes. However, the interaction of PRRSV proteins involved in viral uncoating, such as GP2a, GP3, and GP5, with CD163 is significantly disrupted by the absence of SRCR5. Furthermore, the authors identified calpain 1, a novel host protein involved in PRRSV uncoating. Inhibition of calpain 1 results in the trapping of PRRSV in early endosomes, which are subsequently transported to late endosomes [109].

Given the critical role of SRCR5 in PRRSV infection, researchers have pursued small molecules targeting the CD163 SRCR5 domain as potential antivirals. Using bimolecular fluorescence complementation (BiFC) to assess the protein-protein interactions (PPI) between CD163 and GP2a/GP4, they observed changes in PPI values upon treatment with various compounds, leading to the identification of several small molecules that block CD163-PRRSV interactions and act as potential antiviral candidates [110]. Similarly, He et al. employed this antiviral strategy to screen for nanobodies targeting CD163. Specific nanobodies targeting the SRCR5-9 domains of porcine CD163 were isolated, with seven nanobodies exhibiting strong specificity. Among these, Nb2 demonstrated excellent broad-spectrum inhibitory effects against various PRRSV strains [111]. Before this, a nanobody targeting PRRSV nsp9 named Nb6 was shown to inhibit PRRSV replication in Marc145 cells, though Nb6 could not physically enter cells, highlighting the limitations of such antibodies [112]. With the development of Alphafold 3, molecular docking has enhanced the accuracy and convenience of screening antiviral drugs and antibodies targeting host proteins, offering promising opportunities for the structural-based development of antiviral therapeutics against viral proteins.

PRRSV infection is characterized by persistent viral replication, which represents one of its key features. Following initial infection, the virus spreads to neighboring cells through the release of progeny viral particles, continuing the infection cycle [113]. This process is influenced by various inhibitors and cellular factors, including immune system interference, changes in the viral survival environment, and programmed cell death, such as necrosis and apoptosis. These mechanisms partially suppress PRRSV replication and, in some cases, lead to viral clearance. However, not all PRRSV particles participate in the transmission process. Some viruses are stored within infected cells under conditions of autophagic flux inhibition, maintaining a persistent infection state. Moreover, PRRSV has evolved mechanisms to counteract host immune responses, disrupting multiple signaling pathways to achieve immune evasion. These processes are not isolated but interact to facilitate viral persistence. PRRSV nsp9 is a relatively conserved region in the PRRSV genome, making it a logical viral target for combating PRRSV infection. Liu et al. isolated the nsp9-specific nanobody Nb6 from a phage display library of VHH variable domains, which interacts with the nsp9 encoded by PRRSV [114]. Intracellular expression of Nb6 significantly inhibits PRRSV replication in Marc145 cells. The application of Nb6 in cells is limited as the protein itself cannot physically enter the cells. Wang et al. discovered that fusing Nb6 with the transactivating transcriptional activator (TAT) peptide (TAT-Nb6) facilitated its cellular entry. The TAT peptide delivers Nb6 into Marc145 cells and PAMs in a dose- and time-dependent manner, inhibiting the replication of both genotype 2 and genotype 1 PRRSV strains [112].

PRRSV primarily targets PAMs and the Fc portion of IgG, the most effective phagocytic leukocyte receptor. These phagocytic leukocytes produce various cytokines, chemokines, and lipid mediators to combat viral infections [115]. A study showed that the pig IgG Fc (pFcγ) receptor with the Nb6 chimeric antibody and expressing it in the phage display library system resulted in the Nb6-pFc chimeric antibody. Nb6-pFc enters PAMs via FcγR-mediated endocytosis in a dose-dependent manner. Furthermore, Nb6-pFc exhibits a strong binding affinity for nsp9 and can effectively inhibit PRRSV infection by upregulating proinflammatory cytokine production in PAMs [116].

PRRSV nsp4, a viral 3C-like protein, interferes with the NF-κB signaling pathway by cleaving the essential NF-κB regulatory factors, impairing the initial host antiviral response, and leading to the downregulation of IFN-β expression [117]. As nsp4 is responsible for the process of viral nonstructural proteins, its pivotal role in the PRRSV lifecycle makes it an ideal antiviral target. Liu et al. isolated three nsp4-specific nanobodies, Nb41, Nb42, and Nb43, from a phage display VHH library and developed stable Marc145 cell lines expressing these nanobodies. Nb41 and Nb43 recognized the functional epitopes of nsp4 and effectively inhibited PRRSV replication, protecting Marc145 cells from PRRSV-induced pathogenicity [118].

The PRRSV nucleocapsid protein (PRRSV-N), encoding the ORF7 gene, is relatively conserved, with low mutation rates and good antigenicity and immunogenicity, making it less prone to drug resistance [119]. Duan et al. screened two nsp9-specific nanobodies, PRRSV-RRSV-Nb2. These nanobodies were found to block the binding of the PRRSV-N protein with antibodies from pig serum against PRRSV [120]. The team then evaluated the effects of these nanobodies on PRRSV. The results show that PRRSV-N-Nb1 could block the self-interaction of the N protein after viral assembly, preventing the assembly of viral particles. The key amino acid in the PRRSV-N protein, serine 105, plays an important role in self-interaction with the R97 motif of PRRSV-N-Nb1. In vitro and in vivo studies have demonstrated that PRRSV-N-Nb1 significantly inhibits PRRSV-2 replication [121].

## 8. Summary and Outlook

In recent years, host-derived microRNAs (miRNAs) have emerged as crucial regulators in the intricate interplay between viral pathogens and their host organisms. These small, non-coding RNA molecules, typically 17–25 nucleotides in length, can suppress gene expression by binding to the 3′ untranslated regions (UTRs) of target mRNAs, thereby influencing both viral replication and host immune responses. Studies indicate that certain host-derived microRNAs (miRNAs) exert antagonistic effects on both DNA and RNA viral genomes [122]. Specific miRNAs play a crucial role in determining cellular susceptibility to infection and influencing viral evolution and tissue tropism [123].

In the case of PRRSV, miRNAs such as miR-155, miR-125b, and miR-23 play pivotal roles in altering cellular signaling pathways and promoting antiviral responses. Researchers have identified candidate miRNAs targeting Sn and CD163 receptors. The co-delivery of anti-miRNA (amiRNA) against Sn and CD163 using recombinant adenoviruses (rAd) or exosomes has been shown to significantly reduce PRRSV infection in porcine alveolar macrophages (PAMs). The upregulation of miR-155 suppresses the expression of suppressor of cytokine signaling 1 (SOCS1), a negative regulator of cytokine signaling, leading to enhanced phosphorylation of the signal transducer and activator of transcription 1 (STAT1) and increased IFN-β production, thereby inhibiting viral replication [124]. Similarly, miR-125b has been reported to reduce PRRSV replication by upregulating the NF-κB signaling pathway [125]. Beyond direct targeting of the PRRSV genome, miR-23 enhances IFN-I induction during PRRSV infection by activating interferon regulatory factors 3 and 7 (IRF3/IRF7) [126]. Additionally, miR-181 and let-7f-5p have been identified as potential regulators of key PRRSV entry receptors, including CD163 and MYH9 [127,128]. Studies show that PRRSV infection in PAMs can be partially blocked by antibodies targeting Sn or CD163, while a combination of both antibodies achieves complete inhibition. However, an effective broad-spectrum PRRSV suppression strategy remains elusive. Targeting the essential viral receptor CD163 has emerged as a promising alternative approach for PRRSV inhibition.

Beyond miRNAs, other RNA molecules—including small interfering RNA (siRNA), short hairpin RNA (shRNA), artificial miRNAs, and phosphorodiamidate morpholino oligomers (PMOs)—also exhibit potent antiviral activity against PRRSV [25]. For instance, siRNAs targeting PRRSV nsp1α, nsp9, and ORF7 genes effectively inhibit viral replication in host cells [129]. However, RNA delivery is often constrained by extracellular nuclease degradation, inefficient cellular uptake, or unsuccessful endosomal escape, limiting its full antiviral potential in vivo. To address these challenges, lipid-based formulations such as liposomes or lipid nanoparticles (LNPs) have emerged as the most promising mRNA delivery platforms, offering enhanced stability and targeted delivery for PRRSV vaccine development.

Additionally, modulating viral-encoded IFN antagonists could offer a promising avenue for developing MLVs. Like natural infections, these modified yet live viruses can induce protective humoral and cellular immune responses. However, their lack of some innate immune suppression characteristics of wild-type viruses might enhance the overall immune response and reduce viral attenuation. This strategy has been successfully applied to a variety of viruses, including poxviruses and flaviviruses [130,131].

Ubiquitination plays a crucial role in regulating both the activation and attenuation of innate immune responses to viral infections. Si et al. applied targeted protein degradation technology to develop an attenuated live vaccine, designing a proteolysis-targeting chimera (PROTAC) vaccine by fusing a proteasome-targeting domain (PTD) with influenza virus proteins [132]. This approach harnesses the host cell’s endogenous ubiquitin–proteasome system to degrade host proteins, effectively attenuating the virus and triggering strong, broad-based humoral, mucosal, and cellular immune responses against both homologous and heterologous viral challenges. The team’s latest research provides further evidence of the significant potential for vaccine development with PROTAC technology [133,134]. This strategy can potentially be extended to producing attenuated live vaccines for other pathogens (Figure 4).

One example involves the use of papain-like protease (PLpro), an essential protein in the replication process, and a deubiquitinating enzyme (DUB). Removing the DUB activity of MERS-CoV PLpro in the context of the complete virus generates an attenuated yet replicating virus capable of inducing a strong neutralizing antibody response and providing comprehensive protection against MERS-CoV [135]. This approach could also be adapted for the development of PRRSV MLV vaccines.

## 9. Conclusions

The continuous evolution and genetic diversity of the PRRSV pose substantial challenges for effective control within the swine population. Traditional vaccine platforms, such as KV and MLV, have provided some control but fail to offer broad protection against the diverse and highly recombining PRRSV strains. The recent advent of mRNA-based and subunit vaccines, combined with innovative approaches like PROTAC technology, holds immense potential for enhancing vaccine efficacy. PROTAC technology, which utilizes the host’s ubiquitin–proteasome system to selectively degrade viral proteins, presents a novel strategy for attenuating PRRSV while triggering robust immune responses. This approach offers the advantage of inducing broad immunity by targeting both homologous and heterologous viral strains, making it a promising tool for combating the persistent challenges of PRRSV infection. As PRRSV continues to evolve, the development of adaptive vaccines incorporating PROTAC technology could significantly improve control strategies, safeguarding animal health and ensuring the stability of the global swine industry.

## Figures and Tables

**Figure 1 vaccines-13-00260-f001:**
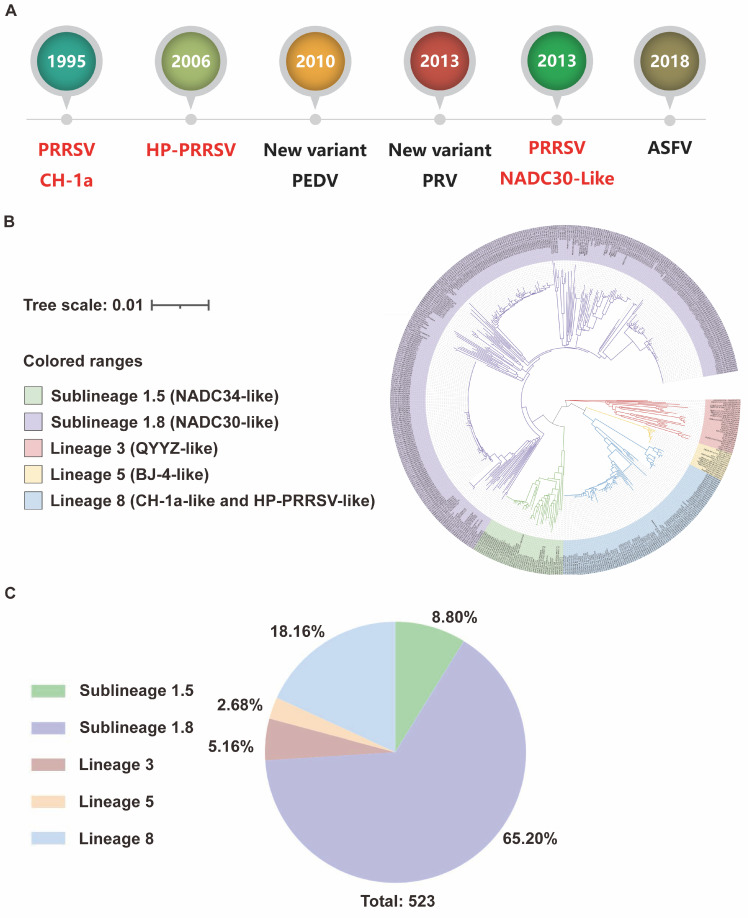
Major epidemics in China’s pig industry over the past 30 years. (**A**) Over the past three decades, China’s pig farming industry has been hit by six major outbreaks of disease. Among them, three outbreaks were closely linked to the porcine reproductive and respiratory syndrome virus (PRRSV). During this period, PRRSV posed a persistent and severe threat to China’s pig farming sector, resulting in significant economic losses. As a result, the development of an effective PRRSV vaccine is crucial not only for controlling the virus but also for ensuring the stability of China’s pig farming industry and safeguarding food security. (**B**,**C**) Phylogenetic analysis based on the ORF5 sequence was conducted to examine the prevalence of PRRSV-2 in China from 2020 to 2024. The results indicate that in recent years, Lineage 1 has been the dominant strain in China, accounting for over 70% of cases. Among these, sublineage 1.8 (NADC30-like) is the most prevalent, representing 65.20%, while sublineage 1.5 (NADC34-like) constitutes 8.80%. Notably, the spread of Lineage 1 strains in China has accelerated, potentially enhancing the virus’s adaptability and immune escape, which could complicate control efforts. Lineage 8 (CH-1a-like and HP-PRRSV-like) strains still account for a significant proportion (18.16%) and maintain a stable presence, with their strong genetic stability warranting continued attention. Meanwhile, Lineage 3 (QYYZ-like) and Lineage 5 (BJ-4-like) strains are less commonly detected (5.16% and 2.68%, respectively), though their potential for evolutionary changes and future spread should not be overlooked.

**Figure 2 vaccines-13-00260-f002:**
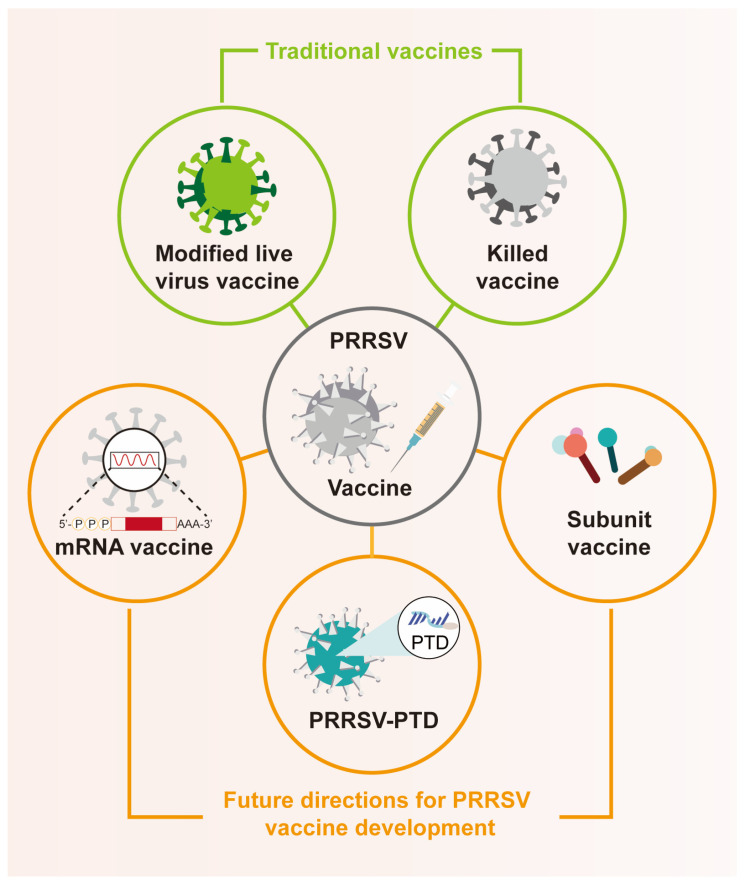
Current types and research directions of PRRSV vaccines. Currently, commercially available PRRSV vaccines primarily include killed vaccines (KV) and modified live vaccines (MLV). While KVs are considered safe, they fail to effectively induce cell-mediated immune (CMI) responses, leading to suboptimal immunity. MLVs, which are more widely used, have limitations such as reversion to virulence, viral recombination, and incomplete protection. In contrast, PRRSV subunit vaccines, based on specific antigens, have garnered attention as potentially safer and more effective alternatives for immunoprotection. Compared to traditional KVs and MLVs, mRNA vaccines offer shorter production timelines, enhanced safety, and longer-lasting efficacy. Additionally, the application of targeted protein degradation technology in the development of attenuated live vaccines has provided significant hope for the comprehensive control of PRRS outbreaks.

**Figure 3 vaccines-13-00260-f003:**
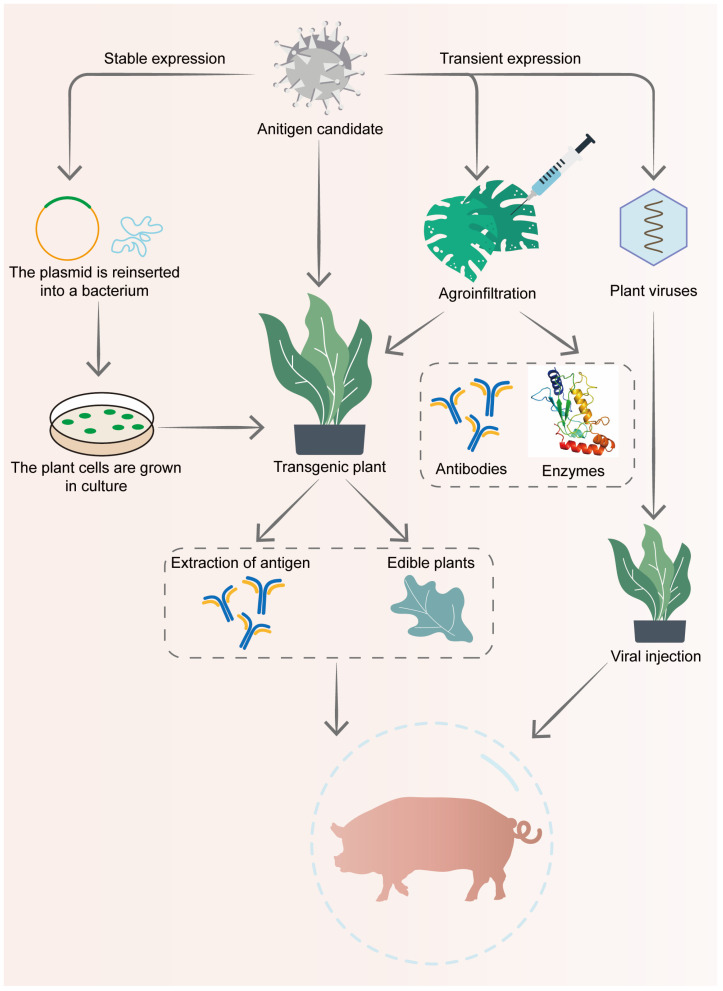
Application of plant-derived vaccines in PRRSV control. Plant-derived vaccines have demonstrated the ability to elicit robust immune responses in both humans and animals. Subunit vaccines produced through plant expression systems offer several advantages, including thermal stability, immunity to contamination by animal pathogens, and cost-effectiveness. In the stable expression route, the antigen gene is inserted into a plasmid, which is then transferred into bacteria. The bacteria infect plant cells, generating transgenic plants capable of long-term antigen production. Moreover, these vaccines can be administered directly through feeding, eliminating the need for purification and processing. The transient expression pathway involves either agroinfiltration, where the antigen gene is directly delivered into plant leaves, or plant viruses, which serve as vectors to transiently express the target gene. These methods enable rapid production of antibodies or enzymes. These attributes make plant-based vaccines a promising tool for PRRSV control.

**Figure 4 vaccines-13-00260-f004:**
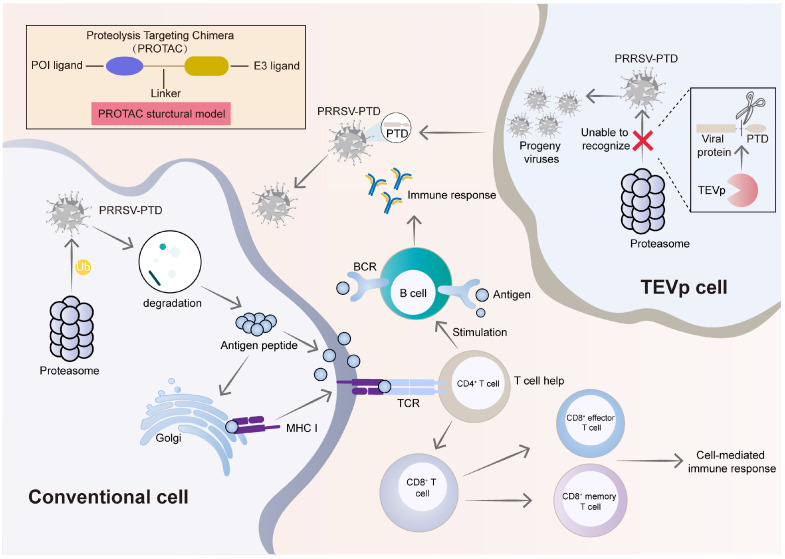
Immune activation mechanisms of the PRRSV-PTD vaccine. The role of PRRSV-PTD (Proteasome-targeting domain, PTD) in enhancing host immune responses and mediating targeted protein degradation, highlighting its potential applications in antiviral strategies and immunotherapy. Within conventional cells, due to the PTD sequence within the genome of PRRSV-PTD, compared with the parental virus, the modified virus undergoes more extensive processing by the host proteasome, resulting in a more abundant and pronounced production of antigenic peptides. These peptides are presented on the cell surface via MHC-I molecules, effectively stimulating CD8+ T cells, which drive more intense cytotoxic responses. Concurrently, PRRSV-PTD-derived antigens activate B cells through their receptors, promoting antibody production and eliciting CD4+ T cell-mediated immune support. On the right, within Tobacco Etch Virus protease (TEVp)-engineered cells, the PTD sequence is specifically recognized and cleaved by intracellular TEVp protein. This cleavage interferes with the proteasomal degradation of viral proteins, allowing the production of progeny PRRSV-PTD virions. The accumulation of PRRSV-PTD subsequently enhances immune responses when introduced into conventional cells. In these cells, PRRSV-PTD triggers robust activation of both cellular and humoral immunity, promoting a strong and coordinated defense against the virus. POI: protein of interest.

**Table 1 vaccines-13-00260-t001:** Statistics of PRRS vaccines used in China.

Source of Vaccine Strain	Name of Vaccine Strain	Vaccine Type	Approval Time
CH-1a	KV CH-1a	KV-Classic strain	2005
M-2	KV M-2	KV-Classic strain	2013
VR-2332	Ingelvac PRRS MLV	MLV-Classic strain	2005
CH-1R	MLV CH-1R	MLV-Classic strain	2007
R98	MLV R98	MLV-Classic strain	2009
HP-PRRSV JXA1	MLV JXA1-R	MLV-Highly pathogenic strain	2011
HP-PRRSV TJ	MLV TJM-F92	MLV-Highly pathogenic strain	2011
HP-PRRSV HuN4	MLV HuN4-F112	MLV-Highly pathogenic strain	2011
HP-PRRSV GD	MLV GDr180	MLV-Highly pathogenic strain	2015
PRRSV SP+GD	MLV PC	Chimeric MLV-Classic plus highly pathogenic	2018

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
