# Peer review of "Porcine Reproductive and Respiratory Syndrome Virus: Challenges and Advances in Vaccine Development"

_vaccines, 2025, doi:10.3390/vaccines13030260_

Round 1

Reviewer 1 Report

Comments and Suggestions for Authors

The authors provide a thorough review of PRRSV vaccines that are now in use and the potential for future vaccines that may improve the immune response to the PRRSV.  The review describes the current advantages and disadvantages of the currently available killed and live vaccines.  

The authors propose that newer technology such as mRNA and SaRNA vaccines may offer distinct advantages to stimulating immunity to PRRSV.  They also describe PROTAC protein vaccines as possible future vaccine technology.  There is some published information on using mRNA as a vaccine for PRRSV but the use of SaRNA and PROTAC are only speculative and there does not appear to be any published information on using these two technologies to develop PRRSV vaccine.  Authors what PRRSV proteins would you target to prepare SaRNA or PROTAC vaccines?

Authors in lines 461-462 you state that "a portion of virus remains latent only reactivated under certain conditions."  I don't believe that there is any evidence the PRRSV establishes a true latency as is the case for some viruses.  Rather it appears that the virus likely continues to replicate at a low level until perhaps immunity wanes and the virus multiplies.    Please review this statement and if you have published evidence of latency please provide.

Reviewer 2 Report

Comments and Suggestions for Authors

My comments focus on the presentation since this is not experimental.

In figure 1 and section 2 and 3, authors  have done a vivid and concise overview on the major pig diseases epidemics in recent 3 decades faced in PRC and commented on the currently commercially available MLVs.

sections 4 to 8 constitutes a  9 consecutive pages of wording which are not joyful to read.  I would encourage authors construct new tables or boxes or figures, listing and commenting on these now developing and promising tools  to aid the understanding for controlling of PRRS. Please try to come up with logics (approaches or mind of thought) or general directions rather than just reciting the experiments published in literature.

Figure 4 is good.

Also consider referring to a recent review which also touches extensively on this issue: Current Status of Porcine Reproductive and Respiratory Syndrome Vaccines, Vaccines 2024, 12, 1387. https://doi.org/10.3390/vaccines12121387

Reviewer 3 Report

Comments and Suggestions for Authors

Peer review:

 Porcine Reproductive and Respiratory Syndrome Virus: Challenges and Advances in Vaccine Development” (vaccines-3438945)

The manuscript presents a review of porcine reproductive and respiratory syndrome virus (PRRSV) vaccines, highlighting recent technological advances in vaccine development and immunization strategies. According to the Abstract of the manuscript, "this review synthesizes and looks ahead to the future directions of vaccine development. It also evaluates the merits and limitations of different PRRSV vaccines, emphasizing the importance of understanding their distinct advantages and drawbacks for devising effective treatment strategies and vaccination programs. The ultimate goal is to mitigate the impact of PRRSV on animal health and the profitability of the swine industry."

In my opinion, the subject is very important. PRRSV is a concerning pathogen which induces pulmonary inflammation and disrupts the host's immune responses, leading to substantial productivity declines and increased susceptibility to secondary infections. I agree with the authors that vaccination is one of the most effective measures for controlling this pathogen infection. The development and deployment of vaccines have been instrumental in reducing the prevalence and impact of infectious diseases in the swine industry.

However, the review is not well-written and presents some structural problems. It is difficult to understand the message... It could not be accepted in this version definitely. First of all, a great effort is needed to create a more fluid text in all sections. There are many sentences in the same paragraph that are not connected, making it difficult to fully understand the content of that paragraph and that section. In addition, the authors need to include some sections, adjust figures and organize better the whole content. I am describing below some aspects that I consider relevant for the review to be publishable in a scientific journal such as Vaccines:

1)    The Abstract does not summarize the article as a whole and should be completely revised. The first two sentences are not related to the topic and take the focus away from the main topic of the article. In addition, it is essential to highlight the topics that will actually be presented in the article. This was only covered in two succinct sentences in the current version (lines 25 to 28). Please expand this part and reduce the others.

2)    The Introduction should be much better written. The text is confusing and very mixed up. It does not allow for a good understanding of the epidemiology, taxonomy and molecular genetic mechanisms of viral evolution. My recommendation is to include specific subsections for the epidemiology and taxonomy of PRRSV. This would help the authors to organize better the information. Please also review the concept of viral genus, species, lineages and strains. I would recommend reading ICTV technical texts.

3)    The Figures are generally good. But Figure 1 is not informative. In addition, some images seem to have been made for children! Remember that the goal is to publish in a scientific journal for research readers!.

4)    The authors do not seem to have had clear criteria for organizing the sections (from 1 to 7) or for defining the title of each one. I recommend a major effort in editing to include informative titles and organize the text within each section.

5)    A review article is a type of manuscript with Discussion in all sections. Therefore avoid a Discussion section. Use “Concluding remarks”, “Perspectives”, etc. But not Discussion!

After these modifications, I think the entire manuscript could be properly evaluated by the reviewers.

Round 2

Reviewer 2 Report

Comments and Suggestions for Authors

The two added tables make it easier to grasp the point authors want to make.

Figure 1B is too small to view. Also slight enlarge Figure 1C.
